# Novel competitive enzyme-linked immunosorbent assay for the detection of the high-risk Human Papillomavirus 18 E6 oncoprotein

**Natalia E. Contreras[1,2], Julieta S. Roldán[1,2], Daniela S. Castillo**[1,2]*

1 Instituto de Investigaciones Biotecnológicas, Universidad Nacional de San Martín (UNSAM)—Consejo Nacional de Investigaciones Científicas y Técnicas (CONICET), San Martín, Buenos Aires, Argentina, 2 Escuela de Bio y Nanotecnologías (EByN), Universidad Nacional de San Martín (UNSAM), San Martín, Buenos Aires, Argentina

* dcastillo@iibintech.com.ar

**Data Availability Statement:** All relevant data are within the manuscript and its Supporting Information files.

## Abstract

Cervical cancer represents a global concern with 604,000 new cases and 342,000 deaths reported annually, with the vast majority diagnosed in low income countries. Despite high-risk Human Papillomavirus (HR HPV)-induced cervical cancer has become highly preventable through prophylactic vaccines, screening programs are critical in the control of cervical carcinogenesis in populations with limited access to vaccination and in older generations of women who have already been exposed to HR HPV infection. The surge of HPV molecular tests has provided a more sensitive and accurate diagnostic alternative to cytology screening. Given that HPV DNA testing presents a low positive predicted value, leading to unnecessary treatment, the E6 oncoprotein from HR HPV types arises as a promising diagnostic marker for its overexpression in transformed HPV-positive cancer cells. For these reasons, this study aimed at obtaining monoclonal antibodies (mAbs) against the E6 oncoprotein of one of the most prevalent HR HPV types worldwide, HPV18, in order to develop a highly specific and sensitive indirect competitive ELISA (icELISA). The production of hybridomas secreting HPV18 E6 mAbs was carried out through a combined tolerization and immunization strategy, in order to avoid cross-reactivity with the E6 protein from low-risk HPV types 6 and 11. We selected the 7D2 hybridoma clone, which recognized HPV18 E6 and showed some cross-reactivity against the HR HPV45 E6 oncoprotein. The 7D2 mAb enabled the development of a sensitive, reliable and reproducible icELISA to detect and quantify small amounts of HPV18 E6 biomarker for cervical cancer progression. The present study establishes a valid 7D2-based icELISA that constitutes a promising bioanalytical method for the early detection and quantification of HPV18 E6 oncoprotein in cervical swab samples and cancer prevention.

**Funding:** The work described in this article was performed with financial support from Agencia Nacional de Promoción Científica y Tecnológica (ANPCyT) [PICT 2019-00036] to DSC. The funders had no role in study design, data collection and analysis, decision to publish, or preparation of the manuscript.

**Competing interests:** The authors have declared that no competing interests exist.

## Introduction

Human papillomavirus (HPV) is a non-enveloped double-stranded circular DNA virus and the most common sexually-transmitted infection [1]. HPVs are classified into five genera (alpha, beta, gamma, mu and nu), based on tissue tropism and subsequent host pathology [2]. The alpha-papillomavirus genera, which infects cutaneous and mucosal epithelia, contains members that are further classified into low-risk (LR) or high-risk (HR) types, depending on their oncogenic ability. The LR viral infections result in benign lesions, and are commonly caused by HPV6 and HPV11; while the HR HPV types are the etiological agents of anogenital and head-and-neck cancers [3]. From these, 14 types have been described (HPV16, 18, 31, 33, 35, 39, 45, 51, 52, 56, 58, 59, 66 and 68), with HPV16 and HPV18 being responsible for 75% of the world's cervical cancer burden [4, 5].

Cervical cancer is the fourth world's most common female malignancy, with the vast majority of cases diagnosed in low resource regions, such as South-Eastern Asia, sub-Saharan Africa and Latin America [6]. The most important risk factor for cervical carcinogenesis is persistent infection with HR HPV types [7]. HR HPV DNA is detected in >99% of all tumors of the uterine cervix [8, 9]. Efficient screening tests are the key to decreasing mortality associated with this disease. Since HPV cannot be cultured from clinical specimens in the laboratory, diagnosis relies on cytologic and molecular methods [1]. Cervical cytology using Papanicolaou (Pap) stained smears has been the standard screening method for detection of premalignant lesions and cervical cancer. Pap tests have successfully reduced cervical cancer incidence and mortality in developing countries, yet they carry a high false-negative rate as well as inter-operator and inter-observer variability [1, 4, 10]. The finding that persistent HR HPV infection has a causal role in the establishment of cervical cancer led to the development of HPV molecular tests. These molecular methods show greater sensitivity and negative predictive value than cytology screening; however, a lower specificity and positive predicted value have also been reported, particularly in young women in whom HPV infection is usually transient [4, 11, 12]. Therefore, detection of HPV DNA through molecular tests do not necessary indicate cervical dysplasia or cancer.

In this scenario, the HR HPV early protein E6 becomes an interesting candidate for the development of cervical cancer screening methods. E6 is a small (~150 amino acids) homodimeric oncoprotein [13], which is overexpressed following HPV invasion into the host cervical cells as episomal DNA or through viral integration into the host's genome [14]. E6 from HR HPV types are functionally different from LR HPV types as they target a plethora of cellular factors, leading to cell cycle progression, evasion of apoptosis, DNA damage and suppression of the host's immune response [3, 14, 15]. Thus, considering that elevated expression of HR HPV E6 is a necessary step for oncogenic transformation of cervical epithelial cells, an assay capable of detecting E6 protein from HR HPV types in cervical swab samples will show a high positive predictive value, serving to accurately identify women at increased risk of cancer progression. For this reason, we aimed at developing an indirect competitive ELISA (icELISA) for the detection of the E6 oncoprotein from one of the most prevalent HR HPV types worldwide: HPV18. In order to obtain highly specific monoclonal antibodies (mAbs) against HPV18 E6 that would not cross-react with the E6 protein from LR HPV types, we have carried out a tolerization procedure with HPV6 and HPV11 E6 proteins before the immunization scheme. With this strategy, we have developed and characterized two mAbs, named 7D2 and 9E2, which bind to HPV18 E6. The 7D2 mAb also shows some cross-reactivity against the HR HPV45 E6 oncoprotein. Using 7D2 mAb, we were able to develop a sensitive, reliable and reproducible icELISA. The 7D2-based icELISA allows the detection of small amounts of E6 oncoprotein in cell extracts from HPV18-positive cervical cancer-derived HeLa cell line and from HPV18 E6

stable HEK293T cells; as well as in HPV-negative cervical cancer-derived C-33 A cell extracts spiked with HPV18 E6 or HPV45 E6 recombinant proteins. The novel developed 7D2 icELISA constitutes a promising analytical method for the early detection of HPV18 E6 oncoprotein in cervical swab samples and cervical cancer prevention.

## Materials and methods

### Production and purification of recombinant HPV E6 proteins

Recombinant *Escherichia coli* expressed-HPV E6 proteins were obtained as N-terminal glutathione S-transferase (GST) and C-terminal 6xHis (His) fusions in pGEX-2T by subcloning the HPV E6-open reading frame followed by a His tag, from pUC57-HPV E6-His constructs (Genscript) into the BamHI/EcoRI sites of the pGEX-2T vector, generating a dual-tagged fusion. The HPV E6 sequences are listed in S1 Table. Constructs for recombinant protein expression and purification were transformed in *E. coli* strain BL21. Cultures transformed with pGEX-2T-HPV E6-His were induced with 1 mM isopropyl β-D-1-thiogalactopyranoside (IPTG) for 18 h at 18˚C. Bacterial cell pellets containing HPV E6 recombinant proteins were resuspended in lysis buffer (50 mM Tris-HCl pH 7.6, 150 mM NaCl, 0.5% Triton X-100, 1 mM PMSF, 2 μg/ml DNase I). After sonication and centrifugation at 10000 rpm for 10 minutes at 4˚C, soluble proteins from the supernatant were purified by immobilized-metal affinity chromatography (IMAC) (S1 Fig). After elution with elution buffer (50 mM Tris-HCl pH 7.6, 500 mM NaCl, 200 mM Imidazole), the most concentrated fractions of HPV E6 recombinant proteins were pooled and dialyzed against dialysis buffer (PBS supplemented with 100 mM NaCl and 5% glycerol) prior to any usage.

### Tolerization, immunization and hybridoma generation

To induce high zone tolerance, 125 ug of purified HPV6 E6-GST and 125 ug of purified HPV11 E6-GST in PBS were injected to 8-week-old male BALB/c mice on day 1 and day 5. For this, the samples were divided into two aliquots, of which two thirds were injected intraperitoneally and the other third was injected subcutaneously into the right and left hind legs. Intraperitoneal immunization with 20 μg of HPV18 E6-GST emulsified in complete Freund's adjuvant was carried out on day 7. Booster injections of 10 μg of HPV18 E6-GST in incomplete Freund's adjuvant were applied 21 and 42 days after the first immunization. Based on the humoral response of a test bleed performed 7 days after the final immunization, the highest BALB/c responder –according to an indirect-enzyme-linked immunosorbent assay (iELISA) (see *indirect-enzyme-linked immunosorbent assay*)– was selected as donor of splenocytes for hybridoma production. Hybridomas were generated by fusion of spleen cells with Sp2/0-Ag14 myeloma cells as described previously [16]. Screening of positive secreting hybridomas was carried out by iELISAs and the selected hybridomas were cloned twice.

### Monoclonal antibodies purification

Monoclonal antibodies were purified by protein G affinity chromatography (mAbia Labs, Argentina) from hybridoma culture supernatants.

### Indirect enzyme-linked immunosorbent assay

Microtiter plates (Nunc Maxisorp 96-well ELISA plates) were coated for 18 h at 4˚C with 100 μl of HPV18 E6-GST (150 ng/well) for hybridoma screening, or with 100 μl of the indicated concentration of recombinant HPV E6 proteins in coating buffer (0.1 M $Na_2HPO_4$ buffer pH 9.5) for cross-reactivity determinations. Following incubation in blocking buffer

(5% skimmed milk in TBS) for 1 h at 37˚C, the plates were further incubated for 1 h at room temperature (RT) with hybridoma supernatants for screening, or with 0.5 μg/ml of 7D2 or 9E2 mAbs in blocking buffer for cross-reactivity assessment. Following four washing steps in TBS containing 0.05% Tween-20, plates were further incubated for 1 h at RT with HRP goat anti-mouse IgG Fcγ fragment specific secondary antibody (Jackson Immunoresearch) at a 1:10000 dilution. Finally, plates were washed four times in TBS with 0.05% Tween-20 and after incubation with the substrate [0.3% $H_2O_2$, 0.1% 3,3',5,5'-tetramethylbenzidine (TMB) in 0.1 M citric acid pH 5] for 10 minutes at RT, the reaction was stopped with 0.2 M $H_2SO_4$. The absorbance at 450 nm was measured with a 800TS microplate reader (BioTek).

### Isotyping of immunoglobulins

The mAb isotype was determined with the Mouse Ig Isotyping Ready-SET-Go kit (Affymetrix, eBioscience) according to the manufacturer's instructions.

### Cell culture

HeLa, SiHa and C-33 A cervical cancer-derived cells, and HEK293T human embryonic kidney cells, were grown in Dulbecco's Modified Eagle Medium (DMEM, Life Technologies) supplemented with 10% fetal bovine serum (Natocor), 50 μg/ml gentamicin sulfate (Sigma-Aldrich), 2 mM GlutaMAX (Gibco) and 1 mM sodium pyruvate at 37˚C in a 5% $CO_2$ humidified atmosphere.

### Western blotting

Cells were lysed with RIPA buffer (50 mM Tris-Cl pH 7.6, 300 mM NaCl, 1% Nonidet P-40, 0.5 mM PMSF). Equal amounts of HPV E6 recombinant proteins or cell lysates were resolved on 10% or 15% SDS-PAGE. After transfer to a nitrocellulose membrane (Hybond-ECL, GE Healthcare), analysis by immunoblotting was performed using 0.35 μg/ml 7D2 or 1.2 μg/ml 9E2 purified monoclonal antibodies, 0.4 μg/ml anti-His tag monoclonal antibody (mAbia labs, Argentina), 1:4000 anti-β-tubulin (Sigma), or 1:10000 anti-GAPDH monoclonal antibody (Santa Cruz), in blocking buffer (1% skimmed milk in TBS). Bound antibodies were recognized with a IRDye 800CW goat anti-mouse IgG secondary antibody (Li-Cor) at a 1:20000 dilution in blocking buffer. The signal was visualized with an Odyssey Infrared Imager (Li-Cor).

### Competitive indirect enzyme-linked immunosorbent assay

Microtiter plates (Nunc Maxisorp 96-well ELISA plates) were coated with 100 μl of HPV18 E6-GST (300 ng/well) in coating buffer (0.1 M $Na_2HPO_4$ buffer pH 9.5) for 18 h at 4˚C. Plates were incubated in blocking buffer (5% skimmed milk in TBS) for 1 h at 37˚C. Recombinant HPV18 E6-GST –serially diluted in TBS– or cell lysates (50 μl), and 50 μl of 0.5 μg/ml 7D2 mAb in blocking buffer, were preincubated for 1 h at RT and then added to the wells. Detection of antibodies with secondary antibody, reaction development for 10 minutes at RT and absorbance measurement were carried out as described in *indirect-enzyme-linked immunosorbent assay*.

### Development of HPV E6 stable cell lines

To obtain stable HPV16 E6 or HPV18 E6-expressing HEK293T cells, we first subcloned the HPV16 E6-His and HPV18 E6-His sequences from pUC57-HPV E6-His constructs (Genscript, described in *production and purification of recombinant HPV E6 proteins*) into the

BamHI/EcoRI sites of the pcDNA3.1 (+) vector (Thermo Fisher plasmid V79020) and, subsequently, into the NheI/EcoRI sites of the lentiviral transfer vector pLB (Addgene plasmid 11619). Lentivirus were generated by cotransfecting the resulting pLB-HPV E6-His vector with the packaging vector pSPAX2 (Addgene plasmid 12260) and the vesicular stomatitis virus (VSV) glycoprotein vector pMD2.G (Addgene plasmid 12259) into HEK293T cells using polyethylenimine (PEI, Polysciences) according to the manufacturer's instructions. Supernatant with lentivirus was collected 3 days after transfection, and low-speed concentration was performed by overnight centrifugation of the viral supernatant at 3000 g and 4˚C. Concentrated viral supernatants were supplemented with 20 mM 4-(2-hydroxyethyl)-1-piperazineethanesulfonic acid (HEPES) pH 7.4 and 12 mg/ml Polybrene (Sigma). For HEK293T transduction, cells were infected with lentivirus by centrifugation at 2500 rpm for 30 minutes at RT. After 4 h of incubation at 37˚C in 5% $CO_2$, inoculum was replaced with complete fresh medium. HPV E6-His expression was evaluated by Immunoblot (as described in *western blotting*).

## HPV typing

Cervical cells from clinical HPV-positive samples were lysed by proteinase K digestion and genomic DNA was extracted by phenol-chloroform-isoamyl alcohol. To confirm the presence of HPV infection, HPV DNA was detected by PCR amplification of a region of ~450 bp in lenght from the highly conserved L1 ORF with the degenerate consensus primers MY09-MY11 [17]. Briefly, after initial denaturation at 95˚C for two minutes, the MY09-MY11 PCR consisted of 35 cycles of denaturation at 95˚C for 35 seconds, annealing at 54˚C for 45 seconds, and extension at 72˚C for 1 minute. A final extension lasted for 5 minutes at 72˚C. Distilled water was used as a negative control of contaminating HPV DNA in reagents.

For HPV typing, Restriction Fragment Length Polymorphism (RFLP) patterns were analyzed (S2 Table) [18]. Aliquots of 4 μl of the crude PCR products were mixed with 2U of the respective restriction enzyme (EcoRI, BamHI, HincII or PstI) and its specific buffer in a total volume of 10 μl. Reactions were incubated for 1 h at 37˚C. The resulting digestion products were detected by 2% agarose gel electrophoresis and visualized by ethidium bromide staining. All extractions, PCRs and digestions were performed in parallel with HeLa and SiHa cervical cancer-derived cell lines.

## Statistical analysis

The software GraphPad Prism 5.0 (GraphPad Software, La Jolla, CA, USA) was used for the non linear fitting of the standard curves to a 4 parameter logistic regression and for the calculation of the IC50 parameter. The CR (cross-reactivity) values were calculated as (IC50 of the antigen for which the mAb was developed/IC50 of the antigen analyzed)*100. Repeatability (intra-plate variability) was assessed by measuring the standard curve six times for the same HPV18 E6 sample on the same ELISA plate. Reproducibility (inter-plate variability) was calculated by measuring the standard curve for different HPV18 E6 samples on two different ELISA plates on different days. The CV (coefficient of variation) was estimated as follows: standard deviation/mean*100.

## Ethical statement

The protocol of animal immunization followed in this study was approved by the Committee on the Ethics of Animal Experiments of the Universidad Nacional de San Martín (Resolution No. 17/2019), according to the recommendations of the Guide for the Care and Use of Laboratory Animals of the National Institutes of Health (NIH). Mice were anesthesized with

isoflurane before sacrifice by cervical dislocation by trained personnel, and all efforts were made to minimize their suffering.

This study is classified as IRB exempted under the NIH guidelines Exemption 4, since it involves the analysis of biospecimens from subjects that cannot be identified. Cervical samples used for matrix interference determination assays came from a panel of 10 cervical swab samples obtained from women with negative cytology. HPV-positive clinical samples were obtained from women with a positive HPV hybrid capture assay, indicating infection with at least one of the following 13 HR HPV types: HPV16, 18, 31, 33, 35, 39, 45, 51, 52, 56, 58, 59, and 68. In all cases, the cervical samples were collected using a cervix brush. Immediately upon collection, cells were pelleted by centrifugation and lysed in RIPA buffer (as described in *western blotting*) for 7D2 icELISA assessment. In addition, genomic DNA from HPV-positive clinical samples was extracted (as described in *HPV typing*) for HPV genotyping.

## Results

### Production and characterization of anti-HPV18 E6 monoclonal antibodies

With the aim to obtain hybridomas secreting antibodies that would specifically recognize HPV18 E6, but would not bind to the LR HPV6 and HPV11 E6 proteins, adult mice were subjected to tolerization prior to the immunization scheme. Immunological tolerance was accomplished by exposing 8-week-old mice to soluble HPV6 E6-GST and HPV11 E6-GST tolerogens in large quantities. A week later, the immunization procedure was started with HPV18 E6-GST antigen. Hybridomas were obtained by fusion of spleen cells from the immunized mice and Sp2/0-Ag14 cells. Screening of positive hybridomas was performed by indirect ELISA (iELISA) with the same antigen used for the immunizations. The 7D2 and 9E2 hybridomas, which expressed high affinity mAbs against HPV18 E6, were selected and cloned twice. We determined that the isotypes of the mAbs were IgG1 for 7D2 and IgG2a for 9E2, both of them containing kappa light chains (Table 1).

Given that the 7D2 and 9E2 mAbs were selected with HPV18 E6-GST recombinant protein, we wanted to confirm that they did not cross react with the LR HPV6 E6-GST and HPV11 E6-GST recombinant proteins. Besides, we decided to study if the developed mAbs were also capable of recognizing E6 oncoproteins of other HR HPV types with high prevalence worldwide (HPV16) and in Latin America (HPV45 and HPV31) [5, 19]. For this purpose, we first evaluated 7D2 and 9E2 mAbs reactivity by Immunoblot assays against HPV E6-GST recombinant proteins from LR (HPV6 and HPV11) and HR (HPV16, HPV18, HPV31 and HPV45) types (Fig 1). Western blot analysis revealed that 7D2 detected HPV18 E6 and, with less sensitivity, HPV45 E6 oncoprotein, while 9E2 only recognized HPV18 E6. Next, to further obtain quantitative data about the ability of the developed mAbs to recognize the E6 oncoprotein from different HPV types, we performed an iELISA with the HPV E6-GST recombinant proteins assayed previously (Fig 2). Both mAbs detected HPV18 E6 with a high sensitivity, which was slightly higher for 7D2. Besides, 7D2 mAb also showed some crossreactivity (10.7%) with HPV45 E6, consistent with the Immunoblot assay (Fig 1). Therefore, these results suggest that 7D2 and 9E2 mAbs are capable of recognizing HPV18 E6; and do not cross-react with the E6 proteins from HPV6 and HPV11 LR HPV types, as well as with the GST tag, indicating a

**Table 1. Isotype of 7D2 and 9E2 mAbs.**

|  | IgG1 | IgG2a | IgG2b | IgG3 | IgA | IgM | Kappa | Lambda |
|---|---|---|---|---|---|---|---|---|
| **7D2** | **3.53** | 0.11 | 0.06 | 0.06 | 0.06 | 0.06 | **3.04** | 0.07 |
| **9E2** | 0.06 | **2.04** | 0.06 | 0.06 | 0.07 | 0.12 | **1.60** | 0.08 |

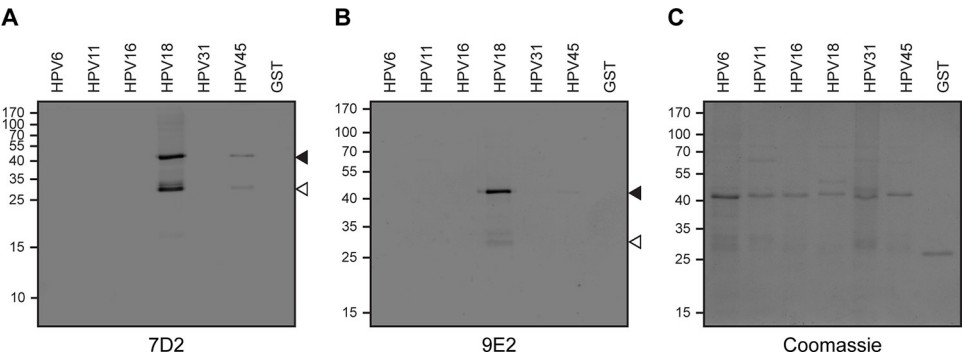

**Fig 1. Western blot analysis of HPV E6 recombinant proteins with 7D2 and 9E2 mAbs.** SDS-PAGE analysis of HPV6, HPV11, HPV16, HPV18, HPV31 and HPV45 E6-GST recombinant protein samples and GST tag alone, analyzed by Immunoblot using **(A)** 7D2 or **(B)** 9E2 mAbs, or by **(C)** Coomassie brilliant blue staining. In **(A,B)** filled arrowheads show HPV E6-GST position, whereas hollow arrowheads show HPV E6-GST cleaved form. The position of the molecular mass standards is indicated on the left.

successfull tolerization procedure. To finally characterize the 7D2 and 9E2 mAbs, we tested their reactivity against the E6 oncoprotein in lysates from HeLa (HPV18-positive) and SiHa (HPV16-positive) cervical cancer-derived cell lines (Fig 3). Wester blot assays showed that both mAbs specifically detect the E6 oncoprotein from HeLa cells, but not that from SiHa cell line. Taken together, these results indicate that 7D2 and 9E2 mAbs are highly specific to a HPV18 E6 linear epitope that is exposed in the native oncoprotein. Given that 7D2 mAb showed a higher sensitivity towards HPV18 E6 and presented some reactivity against HPV45 E6 oncoprotein, we considered it to be a suitable candidate for the development of our icE-LISA immunoassay.

## Development of an indirect competitive enzyme-linked immunosorbent assay using 7D2 monoclonal antibody

The identification of HR HPV E6 oncoproteins as promising diagnostic markers for HPV-driven cervical cancer has drawn the attention of many researchers for the development and validation of E6 screening methods [20–22]. We decided to establish a HPV18 E6 indirect competitive ELISA (icELISA), in which the antigen fixed on the plate and the soluble antigen in the sample compete for binding to the anti-HPV18 E6 7D2 mAb, that is subsequently detected with a HRP-secondary antibody. Hence, a signal decrease indicates the presence of the antigen of interest in the sample analyzed. We established a HPV18 E6-GST concentration of 3 μg/ml fixed on the plate and a preincubation of 0.25 μg/ml 7D2 mAb with the sample for 1 h at room temperature to be the optimal assay conditions. Recombinant HPV18 E6-GST protein was used as the internal standard to develop the HPV18 E6 icELISA. We determined a good correlation to the data ($R^2$ = 0.99) for the 7D2 icELISA standard curve built under the established conditions. The HPV18 E6 concentration that inhibited 50% total binding of the mAb (IC50) in the standard curve of the icELISA was 1247 ng/ml (Fig 4A). Based on a precision profile of the same standards, expressed in terms of the coefficient of variation (CV), we determined that the 7D2-based icELISA had a lower limit of quantification (LLOQ) of 226 ng/ml (Fig 4B) [23]. Next, to evaluate any matrix effects, we measured the HPV18 E6 concentration of cervical cell lysates from women with negative cytology with the 7D2 icELISA. No signal was observed in either blank sample, suggesting that the matrix components do not interfere with the analytical assessment (S3 Table). Finally, the repeatability and reproducibility of the method were determined from several standard curves carried out on the same

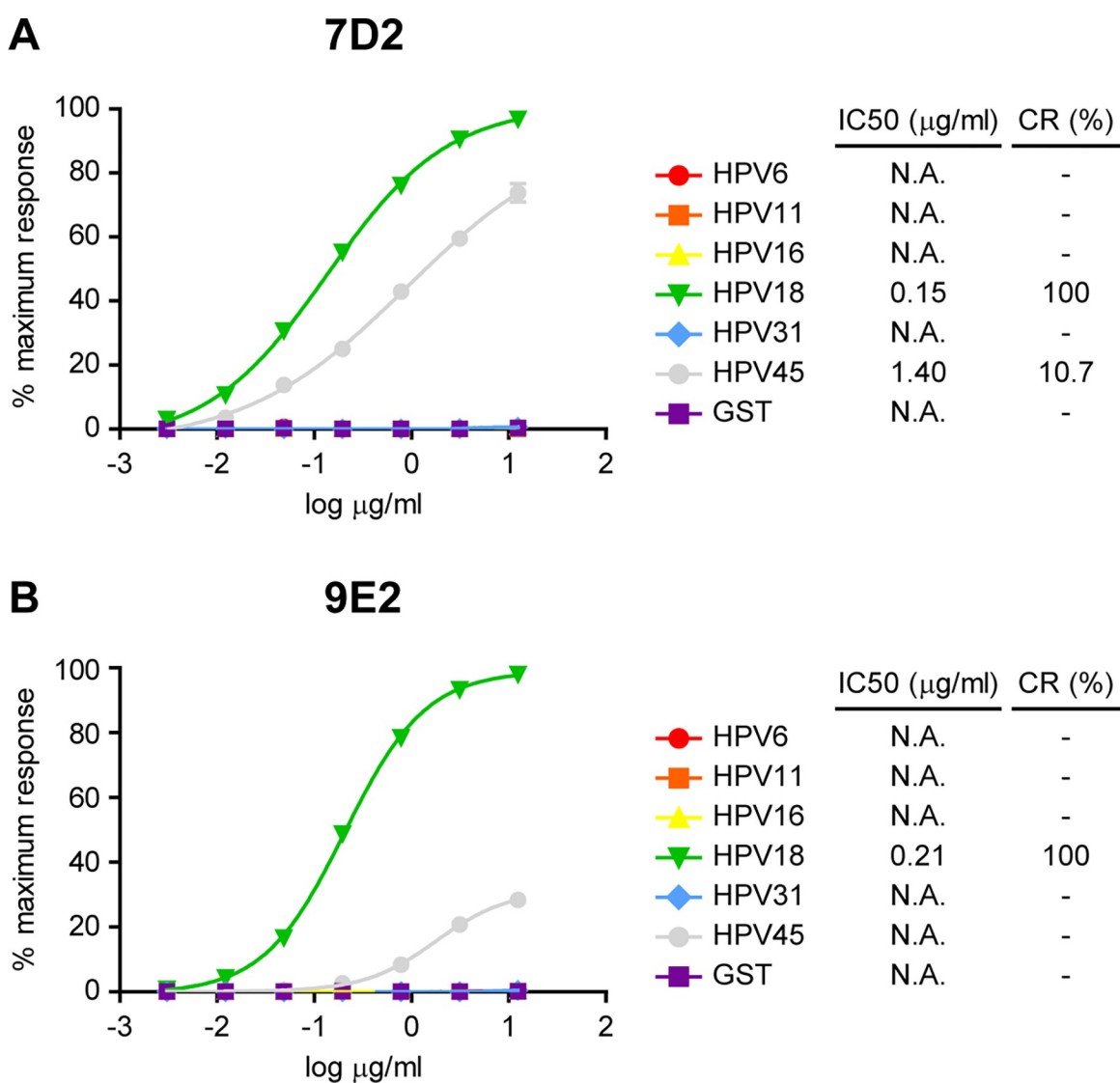

**Fig 2. Comparative reactivity of HPV E6 recombinant proteins by indirect ELISA with 7D2 and 9E2 mAbs.** Standard curves of HPV6, HPV11, HPV16, HPV18, HPV31 and HPV45 E6-GST recombinant protein samples and GST tag alone detected by iELISA with (**A**) 7D2 or (**B**) 9E2 mAbs. Each point of the curve represents mean±SEM of three sample replicates. IC50 and CR values of the mAbs are indicated. N.A.: not applicable.

(intraassay) or on different (interassay) ELISA plates, respectively (Table 2). For the HPV18 E6 standards between 1000 and 8000 ng/ml, we established an intraassay CV of 1.35–13.95% and an interassay CV of 0.72–5.24%. Therefore, our 7D2-based HPV18 E6 icELISA meets the requirements of intraassay and interassay CV used for the validation of bioanalytical methods [24].

To validate the developed 7D2-based icELISA for the detection of HPV18 E6, we performed spike-and-recovery tests. C-33 A cell lysates spiked with low (500 ng/ml), medium (1000 ng/ml) or high (2000 ng/ml) levels of HPV18 E6-GST recombinant oncoprotein were analyzed by the 7D2 icELISA. The average recoveries at the three supplemented levels varied from 93 to 117% and the CV ranged from 0.8 to 7.2% (Table 3). Given that the 7D2 mAb is also capable of binding to some extent to HPV45 E6 oncoprotein (Figs 1 and 2), we carried out the same

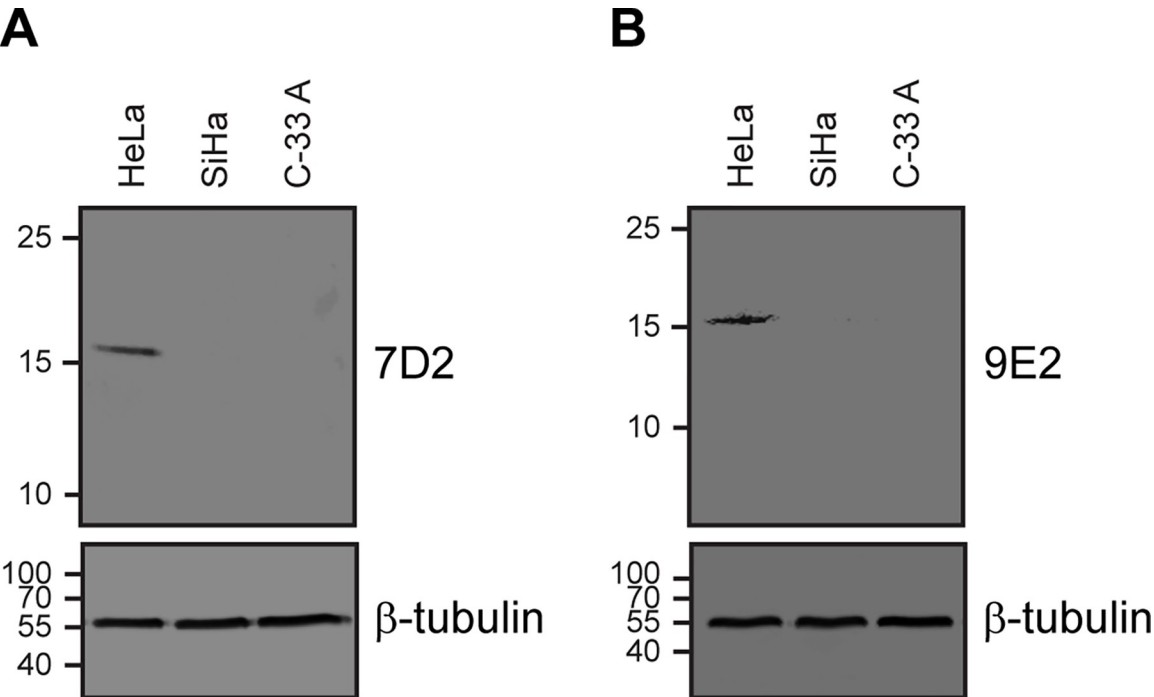

**Fig 3. Cross-reactivity analysis of 7D2 and 9E2 mAbs against E6 oncoproteins from HeLa and SiHa cervical cancer-derived cell lines.** Western blot analysis of HeLa (HPV18+) and SiHa (HPV16+) cell lysates with **(A)** 7D2 or **(B)** 9E2 mAbs. C-33 A HPV-negative cervical cancer-derived cell line was used as control. β-tubulin was used as loading control. The position of the molecular mass standards is indicated on the left.

spike-and-recovery assays with HPV45 E6-GST spiked C-33 A cell samples. We obtained average recoveries from 74 to 124% and CV values from 1.9 to 8.3% for the three supplemented levels (Table 3). Therefore, since all HPV18 E6 recoveries lay within the acceptance range of 80–120% [23], these results act as a proof of concept for the 7D2 icELISA applicability in

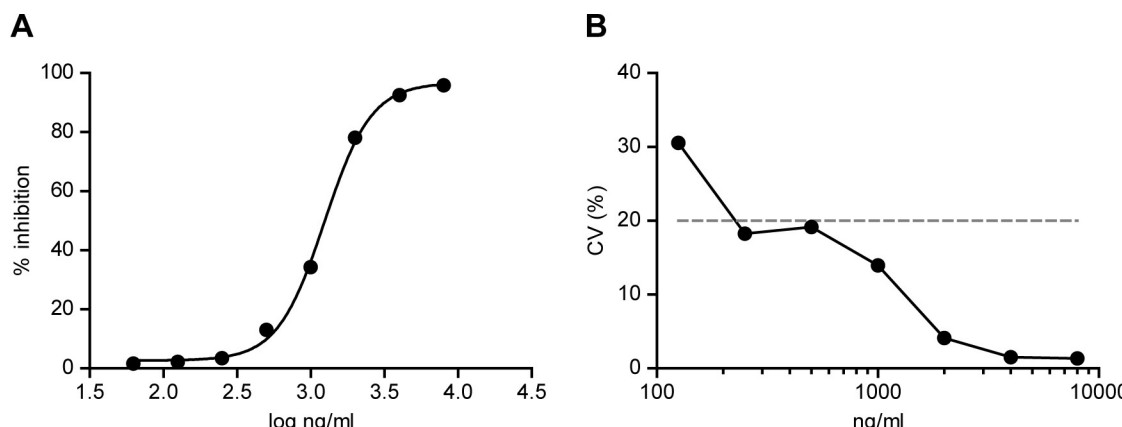

**Fig 4. Standard curve and determination of the lower limit of quantification for the 7D2 indirect competitive ELISA. (A)** Standard curve of the icELISA to detect HPV18 E6 using 7D2 mAb, calculated with a 4 parameter logistic regression fitting with the following equation: $Y = Bottom + (Top-Bottom)/(1+10^{((LogEC50-X)*Hill\ slope)})$, where Bottom = 2.63, Top = 96.73, EC50 = 1241 and Hill slope = 2.73. Each point of the curve represents mean±SEM of six sample replicates. **(B)** Precision profile of standards sextuplicates. The lower limit of quantification (LLOQ) for the determination of HPV18 E6 by the 7D2-based icELISA was determined as the lowest concentration above which the CV < 20%. The dotted line represents the CV = 20% cut-off.

**Table 2. Intraassay and interassay variation of HPV18 E6 standards analyzed by indirect competitive ELISA with 7D2 mAb.**

| HPV18 E6 (ng/ml) | Intraassay (n = 6) | | Interassay (n = 2) | |
|---|---|---|---|---|
| | Inhibition (%)[1] | CV (%) | Inhibition (%)[1] | CV (%) |
| 8000 | 94.94±1.28 | 1.35 | 95.41±0.68 | 0.72 |
| 4000 | 88.76±1.35 | 1.52 | 90.67±2.69 | 2.97 |
| 2000 | 77.31±3.17 | 4.10 | 77.72±0.59 | 0.75 |
| 1000 | 36.92±5.14 | 13.95 | 35.60±1.87 | 5.24 |

[1]Mean±SD

diagnosis. To further validate the developed HPV18 E6 icELISA, we carried out linearity-of-dilution assays with cells extracts from HEK293T cells that stably express HPV16 E6 or HPV18 E6 (S2 Fig). Results showed an average concentration of 4.5 ng/ml of HPV18 E6 oncoprotein per 1000 cells in the dilutions tested (Table 4), indicating the accuracy of the 7D2 icELISA. To finally evaluate the icELISA based on 7D2 mAb for HPV18 E6 detection, we tested cells extracts of HeLa and SiHa HPV-positive cervical cancer-derived cell lines. As expected, lysates from HeLa cells gave high signal (632.5 ng/ml), but no E6 oncoprotein was detected in the extracts from SiHa cells (Table 5). In addition, no signal was observed in HPV16 and HPV31-positive clinical samples (S3 Fig and S4 Table). Overall, these data indicate that the HPV18 E6 icELISA based on 7D2 mAb is sensitive, reliable and reproducible.

## Discussion

The infection with HR HPV is recognized as one of the major causes of infection-associated cancers [25]. Cervical cancer represented the fourth most common cancer in females worldwide in 2020, with 604,000 new cases and 342,000 deaths [6]. The incidence and mortality rates are higher in developing countries, which are burdened with 85% of the cases, principally due to limited access to screening programs and treatment modalities [26, 27]. Cervical cancer is a highly preventable cancer through vaccination and screening. In view that the available HPV vaccines are prophylactic, and do not eliminate persistent HPV infections or interfere with their progression to carcinogenesis, screening programs are crucial in reducing the mortality of this disease [1]. Since the discovery of the association between persistent HR HPV infection and cervical cancer tumorigenesis, many HPV molecular methods have been developed to be performed in parallel of the Pap test. In fact, there is actually a paradigm shift for changing screening from cervical cytology to HPV testing [12]. HPV molecular tests show a higher sensitivity and negative predicted value than Pap tests [1, 4, 10]. The main problem with these molecular screening strategies is that most of them are DNA-based assays, and thus

**Table 3. Recovery analysis of C-33 A cell lysates supplemented with HPV18 E6 or HPV45 E6 by 7D2-based indirect competitive ELISA.**

| HPV-GST | Spike level | Mean±SD (ng/ml)[1] | Recovery (%) | CV (%) |
|---|---|---|---|---|
| HPV18 | 500 ng/ml | 547.5±22.3 | 110 | 4.1 |
| | 1000 ng/ml | 933.2±7.4 | 93 | 0.8 |
| | 2000 ng/ml | 2336.9±168.4 | 117 | 7.2 |
| | 500 ng/ml | 619.1±51.4 | 124 | 8.3 |
| HPV45 | 1000 ng/ml | 898.4±35.4 | 90 | 3.9 |
| | 2000 ng/ml | 1481.7±28.6 | 74 | 1.9 |

[1]Data was obtained from three spiked-sample replicates

**Table 4.** Linearity-of-dilution assays with extracts from HPV16 E6 or HPV18 E6 stable HEK293T cells by indirect competitive ELISA with 7D2 mAb.

| Total cells | Cell line | Mean±SD (ng/ml)[1] | CV (%) | Mean±SD (ng/ml)[1] in 1000 cells |
|---|---|---|---|---|
| **500.000** | HEK293T | n.d. | - | - |
| | HPV16 E6 HEK293T | n.d. | - | - |
| | HPV18 E6 HEK293T | 2442.0±64.6 | 2.6 | 4.88±0.13 |
| **1.000.000** | HEK293T | n.d. | - | - |
| | HPV16 E6 HEK293T | n.d. | - | - |
| | HPV18 E6 HEK293T | 4143.2±224.8 | 5.4 | 4.14±0.22 |

[1]Data was obtained from four sample replicates

n.d.: not detected

deliver a high number of false-positives since the majority of patients will experience a spontaneous infection regression [4, 11, 12]. Only a fraction of HR HPV-positive women will develop high grade lesions and cancer, while most of the infections will remain asymptomatic and will be cleared by the immune system in approximately 12–24 months [3, 28]. Therefore, given that detection of HR HPV DNA in cytological samples does not mandatory indicate cancer progression or disease, HR E6 oncoproteins –which are overexpressed during carcinogenic transformation– have become interesting biomarkers for cervical cancer diagnosis [14, 15]. In view of the need of diagnostic tests to detect HR HPV E6 oncoproteins, we aimed at developing a sensitive and accurate icELISA immunoassay against the E6 oncoprotein from one of the most prevalent HR HPVs worldwide: HPV18.

In this study, we selected and thoroughly characterized 7D2 and 9E2 mAbs against HPV18 E6. These mAbs were designed to detect HPV18 E6, but to do not cross-react with the LR HPV6 and HPV11 E6 proteins. For this purpose, we made use of a tool that enables the generation of discriminatory mAbs to proteins with similar sequence and/or structure: substractive immunization [29, 30]. Given that HR HPV18 and LR HPV6 and HPV11 E6 proteins show sequence homology, we subjected adult mice to tolerization with the LR HPV E6 recombinant proteins prior to challenge with the HPV18 E6 recombinant oncoprotein. It should be noted that this high zone tolerance strategy triggered a tolerization against shared LR and HR HPV epitopes, as well as against the GST tag, allowing a subsequent immunological response directed to the HPV18 E6 desired and specific epitopes. As expected, cross-reaction studies through Western blotting and iELISA indicated that 7D2 and 9E2 mAbs do not recognize LR HPV6 and HPV11 E6 proteins. Both mAbs showed a high reactivity against HPV18 E6 and 7D2 also presented some degree of cross-reativity with the E6 oncoprotein of the third world's most prevalent HR HPV: HPV45 [6]. This result is consitent with the fact that HPV18 and HPV45 E6 oncoproteins exhibit more than 80% of sequence identity, while HPV18 E6's

**Table 5.** HPV18 E6 quantification in lysates from cervical cancer-derived cell lines by 7D2-based indirect competitive ELISA.

| Cell line | HPV18 E6 (ng/ml)[1] |
|---|---|
| **HeLa** | 632.5±153.2 |
| **SiHa** | n.d. |
| **C-33 A** | n.d. |

[1]Mean±SD (n = 2)

n.d.: not detected

sequence homology with the other E6 oncoproteins analyzed –from HR HPV16 and HPV31 types– is ~50% (S5 Table). Finally, in view that 7D2 mAb showed a higher sensitivity towards HPV18 E6 and presented some reactivity against HPV45 E6 oncoprotein, it was considered an appropiate candidate to develop an immunodetection assay to detect HPV18 E6 diagnostic marker.

The enzyme-linked immunosorbent assay (ELISA) is commonly applied for the detection of disease markers in the diagnostic industry due to its relative ease of use, high precission, sensibility and potential for standarization [31]. For this reason, we decided to develop an icELISA based on 7D2 mAb to detect HPV18 E6 oncoprotein in cell extracts. It is worth noting that we have incorporated 1% Nonidet P-40 as the solubilizing agent in the lysis buffer since it is known that this nonionic detergent does not interfere with the antibody-antigen reaction [32]. The 7D2 icELISA showed an IC50 value of 1247 ng/ml, with intraassay and interassay CV values lower than 20%, indicating its repeatability and reproducibility [24]. In addition, we obtained a LLOQ of 226 ng/ml. The results of the spike-and-recovery tests indicated that the icELISA based on 7D2 mAb was sensitive, reliable and reproducible. The average HPV18 E6 recoveries were between the ideal range from 80 to 120% [23], and CV values were lower than 10%. It is important to highlight that the developed icELISA presented an acceptable recovery of C-33 A cell lysates that were formerly spiked with HPV45 E6 recombinant oncoprotein, considering that 7D2 mAb exhibited only 10.7% of cross-reactivity against this HR HPV E6 protein by iELISA. Furthermore, we also demonstrated the accuracy of the 7D2 icELISA with extracts from HPV18 E6 stable HEK293T cells by linearity-of-dilution assays. Finally, we validated the icELISA based on 7D2 mAb measuring the concentration of HPV18 E6 from lysates of HeLa HPV18-positive cervical cancer-derived cells, confirming the assay's specificity as it did not detect any E6 neither from SiHa HPV16-positive cell line nor from HPV16 and HPV31-positive clinical samples.

In conclusion, a highly specific mAb to HPV18 E6 was selected and characterized. It enabled the development of an accurate and sensitive icELISA, which stands as an encouraging alternative to existing molecular screening methods. In spite of the availability of prophylactic vaccines which protect against infection with oncogenic HPV types, HPV-linked cervical cancer is expected to remain a serious public health problem for decades, particularly in low resource regions, thus emphasizing the importance of the accessibility to reliable screening methods [1, 26, 27]. The present study establishes a valid 7D2 icELISA, which constitutes a promising bioanalytical method and a boost to continue developing HR HPV E6 oncoprotein-based screening strategies for the prevention of cervical cancer.

## Supporting information

**S1 Fig. Purification of HPV E6 recombinant proteins.** SDS-PAGE analysis of **(A)** HPV6, **(B)** HPV11, **(C)** HPV16, **(D)** HPV18, **(E)** HPV31 and **(F)** HPV45 E6 dual-tagged (GST/His) recombinant protein fractions from the IMAC purification, analyzed by Coomassie brilliant blue staining. The position of the molecular mass standards is indicated on the left. Arrowheads indicate the HPV E6 recombinant proteins. FT: flow through; F: fraction.
(TIF)

**S2 Fig. Development of stable HPV16 E6 and HPV18 E6-expressing HEK293T cells.** Western blot analysis of cell extracts from HPV16 E6-His and HPV18 E6-His HEK293T cells with anti-His monoclonal antibody. Untransduced HEK293T cells were used as control. GAPDH was used as loading control. The position of the molecular mass standards is indicated on the left.
(TIF)

**S3 Fig. HPV typing of HPV-positive clinical samples. (A)** Amplification by PCR with the consensus primers MY09-MY11 for the L1 gene. The negative control corresponds to molecular grade water. **(B)** RFLP pattern of L1 PCR product with EcoRI, BamHI, HincII and PstI. The upper pannel corresponds to patient 11 (P11) and the lower pannel corresponds to patient 12 (P12), which were characterized as HPV16 and HPV31-positive, respectively. **(C)** Digestion control of HincII restriction enzyme by pLB vector linearization under the same digestion conditions as L1 PCR products in **(B)**. In **(A)** and **(B)** HeLa and SiHa cervical cancer-derived cell lines were used as HPV18 and HPV16-positive controls, respectively. In **(C)** the filled arrowhead shows the open circular pLB conformation, whereas the hollow arrowhead shows the linear pLB form.
(TIF)

**S1 Table. LR and HR HPV E6 sequences subcloned into the pUC57 vector.**
(DOCX)

**S2 Table. Restriction fragment length polymorphism (RFLP) patterns for HR HPVs.** Patterns were predicted from DNA sequences of MY09-MY11 L1 PCR products.
(DOCX)

**S3 Table. Assessment of cervical cell lysates matrix effects for the 7D2 indirect competitive ELISA.** The concentration of HPV18 E6 of 10 different cervical cell lysates from women with normal cytology (blank samples) was calculated by the 7D2-based icELISA.
(DOCX)

**S4 Table. HPV18 E6 quantification in lysates from HPV-positive clinical samples by 7D2-based indirect competitive ELISA.** The concentration of HPV18 E6 from HPV16 and HPV31-positive clinical samples was calculated by the 7D2 icELISA.
(DOCX)

**S5 Table. Sequence identity matrix of HPV16, HPV18, HPV31 and HPV45 E6 oncogenes.**
(DOCX)

**S1 Raw images.**
(TIF)

**S2 Raw images.**
(TIF)

**S3 Raw images.**
(TIF)

**S4 Raw images.**
(TIF)

**S5 Raw images.**
(TIF)

## Acknowledgments

We are thankful to Roberto L. Hurtado, MD, and Romina Gimenez, MD, for providing the cervical samples. DSC is a member of the Research Career of CONICET.

## Author Contributions

**Conceptualization:** Daniela S. Castillo.

**Data curation:** Daniela S. Castillo.

**Formal analysis:** Natalia E. Contreras, Julieta S. Roldán, Daniela S. Castillo.

**Funding acquisition:** Daniela S. Castillo.

**Investigation:** Natalia E. Contreras, Julieta S. Roldán, Daniela S. Castillo.

**Methodology:** Natalia E. Contreras, Julieta S. Roldán, Daniela S. Castillo.

**Project administration:** Daniela S. Castillo.

**Supervision:** Daniela S. Castillo.

**Validation:** Natalia E. Contreras, Julieta S. Roldán, Daniela S. Castillo.

**Visualization:** Daniela S. Castillo.

**Writing – original draft:** Daniela S. Castillo.

**Writing – review & editing:** Natalia E. Contreras, Julieta S. Roldán, Daniela S. Castillo.

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
