## [Decision Letter · Decision Letter 0]

3 Apr 2023

PONE-D-23-06213Novel Competitive Enzyme-Linked Immunosorbent Assay for the detection of the high-risk Human Papillomavirus 18 E6 oncoproteinPLOS ONE

Dear Dr. Castillo,

Thank you for submitting your manuscript to PLOS ONE. After careful consideration, we feel that it has merit but does not fully meet PLOS ONE’s publication criteria as it currently stands. Therefore, we invite you to submit a revised version of the manuscript that addresses the points raised during the review process. Your manuscript has been favorably received by all three independent reviewers. However, they have suggested some revisions to improve the manuscript.

We look forward to receiving your revised manuscript.

Kind regards,

Arunava Roy, Ph.D.

Academic Editor

PLOS ONE

Journal Requirements:

3. Regarding blot/gel data: PLOS ONE now requires that submissions reporting blots or gels include original, uncropped blot/gel image data as a supplement or in a public repository. This is in addition to complying with our image preparation guidelines described at https://journals.plos.org/plosone/s/figures#loc-blot-and-gel-reporting-requirements. These requirements apply both to the main figures and to cropped blot/gel images included in Supporting Information. If the manuscript is positively reviewed, we will ask the authors to provide any missing raw image data for blot/gel results when they submit their first revision. As part of your review, please ensure that figures reporting blot or gel images comply with the journal’s image preparation guidelines and that the original data are provided following the journal’s request.  If you have any questions or concerns about blot/gel figures or data for this submission, please email us at plosone@plos.org before issuing a decision letter

Reviewers' comments:

Reviewer's Responses to Questions

**Comments to the Author**

1. Is the manuscript technically sound, and do the data support the conclusions?

Reviewer #1: Partly

Reviewer #2: Yes

Reviewer #3: Partly

2. Has the statistical analysis been performed appropriately and rigorously? 

Reviewer #1: Yes

Reviewer #2: Yes

Reviewer #3: Yes

3. Have the authors made all data underlying the findings in their manuscript fully available?

Reviewer #1: Yes

Reviewer #2: Yes

Reviewer #3: Yes

4. Is the manuscript presented in an intelligible fashion and written in standard English?

Reviewer #1: Yes

Reviewer #2: Yes

Reviewer #3: Yes

5. Review Comments to the Author

Reviewer #1: In this manuscript, Contreras N.E. et al aimed at developing a icELISA to detect high risk HPV18 E6 oncoprotein. This study overall is well designed and significant to improve the diagnostic accuracy of cervical cancer. But the assay developed here can be better supported with more controls and additional experiments to be proved to be a promising novel bioanalytic method for the early detection and quantification HR HPV E6 oncoprotein. Details are listed below.

1. For the first part in the results session that identified the 7D2 mAb, it is not clear to me that if you have also identified other mAb that might be able to detect more of the HR HPV strains but not the LR strains. Is there any other antibodies that you obtain can react with all HR strain without reacting with all LR? If not, it would be more convincing if you include data from some other antibodies to that you obtain in your screening to compare with the 7D2 mAb, like the reactivity assay, wb analysis etc together with 7D2 mAb. In this way you can better prove that this 7D2 mAb is the best you get because it can at least detect the most prevalent HR HPV strain.

2. At the beginning of the second part where you claimed you established a HPV18 E6-GST concentration of 3 μg/ml fixed on the plate and a preincubation of 0.25 μg/ml 7D2 mAb with the sample for 1 h at room temperature to be the optimal assay conditions, could you please provide your data to show how the optimal concentration of E6-GST and the 7D2 mAb as well as the incubation time and temperature was determined? Not clear to me how you established and determined these variables.

3. You have used cervical cell lysates from women with negative cytology in S2 table to show your assay did not detect any signal from no cervical cancer samples, but in the whole paper you have not used any positive cancer clinical sample to test your assay, only demonstrated cell line results. I do think it is important to test in some positive clinical samples to prove your assay is capable and applicable for detecting E6 oncoprotein from clinical cervical cell lysates.

4. For the HEK293T cells data you showed in table 4, it would be better to include the HPV45 E6 expressing cells. But this is a minor point.

Reviewer #2: The manuscript entitled "Novel Competitive Enzyme-Linked Immunosorbent Assay for the detection of the high-risk Human Papillomavirus 18 E6 oncoprotein" was done by Contreras et.al. focused on identifying a monoclonal antibody against the E6 protein which is overexpressed in the transformed HPV-positive cancer cells. The authors were able to develop an ELISA assay that was highly specific and sensitive toward the E6 protein from the HPV18 strain. The authors gave great consideration to the point that the monoclonal antibodies should not be cross-reactive in the low-risk HPV types. The authors proposed using this as a promising method for detection of the HPV18.

The rationale for the authors to generate such an antibody against HPV18 was due to the fact that there is a very low positive predictive value, especially using cytological testing and molecular diagnosis. Due to this, there is an increase in unnecessary treatment of individuals. To overcome this the authors decided to develop a highly specific and sensitive detection method using indirect competitive ELISA using a monoclonal antibody. To generate the antibodies highly specific towards the E6 protein of the HPV18 they used a tolerization strategy.

The authors had a clear hypothesis and the approach used for testing the hypothesis is streamlined by a set experimental design. The quality of data presented and the statistical test are done appropriately to the best of my knowledge. The materials and methods section is complete and gives all the necessary information. Finally, the discussion is well-written and proposes further studies. However, there are some concerns which are as follows.

Is the protein HPV18 E6-GST mentioned on line 154 somewhat different than the same mentioned on line 157 which is mentioned as “oncoprotein”? If so what is the difference if not kindly put the same name everywhere to avoid confusion?

For the indirect competitive ELISA, what is the HRP labeled to? the HPV E6 or the antibody?

Can the authors show the competitive ELISA with direct labeling of the HRP to the antigen or the antibody instead of indirect competitive ELISA?

For the Indirect ELISA on line 126, how was the monoclonal antibody already used when this particular EILSA was used for screening the mice with the best immunization? The authors should explain this.

Can the authors test the iELISA with some of the clinical samples?

Overall the work done by Contreras et.al. is commendable and adds to the necessary information.

Reviewer #3: The manuscript titled “Novel Competitive Enzyme-Linked Immunosorbent Assay for the detection of the high-risk Human Papillomavirus 18 E6 oncoprotein” submitted to Plos one, describes a new ELISA technique developed by the authors Natalia E. Contreras, Julieta S. Roldán and Daniela S. Castillo. I appreciate the efforts undertaken by the authors in developing a novel ELISA based assay to detect E6 oncoprotein. It would be of diagnostic relevance if E6 oncoprotein could be detected directly from clinical specimen. I would like to recommend a couple of additional experiments (comments 1&2), an additional table (comment 3) and a few minor corrections (Comments 4-8), which I believe will substantially improve the quality of the article.

1) The test needs to be clinically validated before it can be recommended as screening tool. As a preliminary step authors could test their assay on at least a few clinically confirmed cytology positive/cervical cancer samples

2) HPV DNA PCR test is considered the gold standard test for HPV detection. I would recommend performing a comparative analysis of results from both PCR and this new ELISA test on confirmed cervical cancer biopsy samples. This would be highly informative as it would give out a direct comparison of accuracy, sensitivity and specificity of new method as against the current gold standard.

3) Include a table with sensitivity and specificity of this new test.

4) Lines 265-267: Please restructure the sentence (“The standard curve for the …..” ) for better clarity. The sentence mentions “good correlation to the data”- which data are the authors referring to?

5) Supplementary table 3 needs to be restructured: The sequence identity matrix several duplicate values- eg: either the last 2 rows or last 2 columns could be deleted without loss of any information; even then the value 55.6 (HPV 18) remains repeated.

6) Line 102: Word “dialysis” misspelt

7) Line 131: “Tetramethylbenzidine” misspelt

8) Line 162: Word “subcloned” misspelt

6. PLOS authors have the option to publish the peer review history of their article (what does this mean?). If published, this will include your full peer review and any attached files.

Reviewer #1: No

Reviewer #2: No

Reviewer #3: **Yes: **Priya Ramesh Prabhu

---

## [Author Response · Author response to Decision Letter 0]

17 Jul 2023

Response to Journal Requirements:

We have made all necessary changes to meet PLOS ONE's style requirements for submission.

We have provided the additional information regarding the experiments involving animals in the Materials & Methods section.

3. Regarding blot/gel data: PLOS ONE now requires that submissions reporting blots or gels include original, uncropped blot/gel image data as a supplement or in a public repository. This is in addition to complying with our image preparation guidelines described at https://journals.plos.org/plosone/s/figures#loc-blot-and-gel-reporting-requirements. These requirements apply both to the main figures and to cropped blot/gel images included in Supporting Information. If the manuscript is positively reviewed, we will ask the authors to provide any missing raw image data for blot/gel results when they submit their first revision. As part of your review, please ensure that figures reporting blot or gel images comply with the journal’s image preparation guidelines and that the original data are provided following the journal’s request. If you have any questions or concerns about blot/gel figures or data for this submission, please email us at plosone@plos.org before issuing a decision letter.

The original blot or gel results are provided as S1, S2, S3, S4 and S5 raw images.tif

Response to Reviewers' comments:

Reviewer #1: In this manuscript, Contreras N.E. et al aimed at developing a icELISA to detect high risk HPV18 E6 oncoprotein. This study overall is well designed and significant to improve the diagnostic accuracy of cervical cancer. But the assay developed here can be better supported with more controls and additional experiments to be proved to be a promising novel bioanalytic method for the early detection and quantification HR HPV E6 oncoprotein. Details are listed below.

1. For the first part in the results session that identified the 7D2 mAb, it is not clear to me that if you have also identified other mAb that might be able to detect more of the HR HPV strains but not the LR strains. Is there any other antibodies that you obtain can react with all HR strain without reacting with all LR? If not, it would be more convincing if you include data from some other antibodies to that you obtain in your screening to compare with the 7D2 mAb, like the reactivity assay, wb analysis etc together with 7D2 mAb. In this way you can better prove that this 7D2 mAb is the best you get because it can at least detect the most prevalent HR HPV strain.

As it was suggested by Reviewer #1, we included the characterization of 9E2 mAb, which was also obtained from our screening. These results correspond to Table 1 (9E2 mAb isotype determination), Fig. 1B (9E2 mAb cross-reactivity assessment by Western Blot), Fig. 2B (9E2 mAb cross-reactivity evaluation by iELISA) and Fig. 3B (9E2 mAb cross-reactivity analysis against E6 oncoproteins from HeLa and SiHa cervical cancer-derived cells lines). Given that 9E2 mAb only detected E6 oncoprotein from HPV18, but with less sensitivity than 7E2 mAb, we decided to continue working with the latter. The 7D2 mAb was chosen as an appropriate candidate to develop the icELISA for its higher sensitivity against HPV18 E6 oncoprotein, as well as for its ability to also detect HPV45 oncoprotein. 

2. At the beginning of the second part where you claimed you established a HPV18 E6-GST concentration of 3 μg/ml fixed on the plate and a preincubation of 0.25 μg/ml 7D2 mAb with the sample for 1 h at room temperature to be the optimal assay conditions, could you please provide your data to show how the optimal concentration of E6-GST and the 7D2 mAb as well as the incubation time and temperature was determined? Not clear to me how you established and determined these variables.

Development of ELISA assays imply the establishment of the ideal concentrations of each assay reagent, as well as temperature and incubation times. Optimization steps aim at achieving the best signal:noise ratio, which is the signal generated by a sample containing analyte relative to the signal of the same sample without analyte. Although each component of an ELISA is described separately, in many instances it is possible to optimize two components simultaneously by performing a checkerboard titration or grid experiment. This design permits the analysis of different concentrations of two reagents in each well to obtain the best signal:noise ratio. Hence, to establish the optimal HPV18 E6-GST concentration fixed on the plate and the 7D2 mAb preincubation concentration for our icELISA, we carried out a two-dimensional serial dilution experiment (see Table 1 from the "Response to reviewers" file). It is usually advised to stop the TMB reaction with a signal:noise ratio of at least ten, and when the maximum O.D. 450 nm is between 1.0 and 2.0 O.D. units. From our experience, we consider an O.D. 450 nm equal or greater than 1.2 O.D. units as a satisfactory cut-off value for our B0 (B0 is the standard which contains no competing antigen (HPV18 E6-GST) and corresponds to the maximum O.D. 450 nm measured). Therefore, considering these optimization criteria, we established a HPV18 E6-GST concentration of 3 μg/ml fixed on the plate and a preincubation of 0.25 μg/ml 7D2 mAb with the sample as the optimal assay concentrations. Regarding the temperature and time variables, we determined that 1 h at room temperature (RT) was acceptable for this step. The timing of an assay is typically the balance between keeping the procedure as short as possible and the need to reach equilibrium in order to achieve maximum sensitivity and specificity. We have already developed two icELISA assays (https://doi.org/10.1371/journal.pone.0182447 and https://doi.org/10.1371/journal.pone. 0256220), and the timing and temperature variables which have been already established for each step were maintained for the 7D2 icELISA: blocking for 1 h at 37°C; sample and mAb preincubation for 1 h at RT; sample and mAb incubation in the plate for 1 h at RT; and secondary antibody incubation for 1 h at RT.

3. You have used cervical cell lysates from women with negative cytology in S2 table to show your assay did not detect any signal from no cervical cancer samples, but in the whole paper you have not used any positive cancer clinical sample to test your assay, only demonstrated cell line results. I do think it is important to test in some positive clinical samples to prove your assay is capable and applicable for detecting E6 oncoprotein from clinical cervical cell lysates.

As in most developing countries, in Argentina, the standard procedure for cervical cancer screening consists of cytology and/or colposcopy determinations. HPV molecular tests are advised when the cytological or colposcopy exam has yielded an abnormal result. This kind of molecular tests aren't usually covered by the patient's health insurance and, in that case, are only performed if the patient agrees to bear the costs associated with the test. The routine HPV molecular assay is the hybrid capture test, which reports positive detection of at least one type of the following oncogenic HR HPVs: HPV16, 18, 31, 33, 35, 39, 45, 51, 52, 56, 58, 59 and 68. Therefore, this test does not distinguish which HPV genotype is present in the sample.

To collect HPV-positive clinical samples, we had to establish contact with several Gynecology Departments from Medical Centers, so that they could provide us with swab samples from women who have had a positive result from the hybrid capture test, and had an upcoming visit scheduled at the doctor's office. Despite all our efforts, we were able to collect very few samples. Besides, the biological sample (endocervical cells) had to be large enough for the determination of the presence of HPV18 E6 by the 7D2 icELISA and, in parallel, for the genomic DNA extraction for HPV genotyping. We were able to establish the genotype from two samples, which were characterized as HPV16 and HPV31 by PCR (MY09-MY11) followed by RFLP. Both samples showed no signal in our 7D2-based icELISA as expected, indicating the assay's specificity as it did not detect the E6 oncoprotein from other HR HPV types than HPV18 (and HPV45). These results correspond to S3 Fig and S4 Table from the revised manuscript. Unluckily, we weren't able to collect any HPV18-positive clinical sample to test the 7D2 icELISA. We intend to further validate our immunoassay with a larger number of clinical samples. Yet, this future work needs the gathering of a high number of swab specimens and hence a broader collaboration with hospital centers. Collaborative projects between the academia and health centers take time and administrative paperwork, which are significant barriers for research. Nonetheless, it should be emphasized that our manuscript entails a novel immunoassay for the detection and quantification of the E6 oncoprotein from one of the most prevalent HR HPV types worldwide. No other similar studies have been reported about this topic up to date. Our work aims to act as a proof of concept of the capability of an indirect competitive ELISA to detect and quantify HPV18 E6 and, to some extent, HPV45 E6 oncoprotein. We have validated our immunoassay in all the possible ways we could in our laboratory, stating it as a promising screening method for cervical cancer prevention.

4. For the HEK293T cells data you showed in table 4, it would be better to include the HPV45 E6 expressing cells. But this is a minor point.

We agree with Reviewer #1 that it would have been interesting to include HPV45 E6 expressing cells in our study. Given that obtaining stable cell lines is cost and time consuming, we focused our efforts in the development of HPV16 E6 and HPV18 E6 stable cell lines since these HR HPV types are the most prevalent worldwide.

Reviewer #2: The manuscript entitled "Novel Competitive Enzyme-Linked Immunosorbent Assay for the detection of the high-risk Human Papillomavirus 18 E6 oncoprotein" was done by Contreras et.al. focused on identifying a monoclonal antibody against the E6 protein which is overexpressed in the transformed HPV-positive cancer cells. The authors were able to develop an ELISA assay that was highly specific and sensitive toward the E6 protein from the HPV18 strain. The authors gave great consideration to the point that the monoclonal antibodies should not be cross-reactive in the low-risk HPV types. The authors proposed using this as a promising method for detection of the HPV18.

 The rationale for the authors to generate such an antibody against HPV18 was due to the fact that there is a very low positive predictive value, especially using cytological testing and molecular diagnosis. Due to this, there is an increase in unnecessary treatment of individuals. To overcome this the authors decided to develop a highly specific and sensitive detection method using indirect competitive ELISA using a monoclonal antibody. To generate the antibodies highly specific towards the E6 protein of the HPV18 they used a tolerization strategy.

 The authors had a clear hypothesis and the approach used for testing the hypothesis is streamlined by a set experimental design. The quality of data presented and the statistical test are done appropriately to the best of my knowledge. The materials and methods section is complete and gives all the necessary information. Finally, the discussion is well-written and proposes further studies. However, there are some concerns which are as follows.

Is the protein HPV18 E6-GST mentioned on line 154 somewhat different than the same mentioned on line 157 which is mentioned as “oncoprotein”? If so what is the difference if not kindly put the same name everywhere to avoid confusion?

We have changed this phrase to avoid confusion (lines 156-158 from the revised manuscript).

For the indirect competitive ELISA, what is the HRP labeled to? the HPV E6 or the antibody?

Can the authors show the competitive ELISA with direct labeling of the HRP to the antigen or the antibody instead of indirect competitive ELISA?

In our 7D2-based icELISA, the HPV18 E6-GST antigen fixed on the plate, and the soluble HPV18 E6 oncoprotein in the standard/sample complete for binding to the anti-HPV18 E6 7D2 mAb, which is subsequently detected with a HRP-secondary antibody. As the concentration of the antigen in the standard/sample increases, the amount of available 7D2 mAb to bind to the HPV18 E6-GST antigen fixed on the plate decreases. Therefore, there is an inverse relationship between the O.D. and the amount of antigen in the standard/sample. 

As the reviewer suggests, we could have established a direct competitive ELISA, labeling our 7D2 mAb to HRP. Anyway, to increase the sensitivity of our assay, we preferred to develop an indirect competitive ELISA, using an anti-mouse HRP-secondary antibody, in order to amplify the signal.

For the Indirect ELISA on line 126, how was the monoclonal antibody already used when this particular EILSA was used for screening the mice with the best immunization? The authors should explain this.

In the Materials & Methods section, we described the protocol for our indirect ELISA. We stated that the hybridoma supernatants were used as the source of primary antibodies for screening. On the other hand, once we selected and cloned twice the desired hybridomas, we started their purification and characterization. To test their cross-reactivity against the E6-GST recombinant proteins, we performed indirect ELISA assays, incubating with 7D2 or 9E2 primary mAbs. To avoid any misunderstanding, we corrected the protocol in the Materials & Methods section, describing the use of hybridoma supernatants or purified mAbs for each procedure (lines 125-127 from the revised manuscript).

Can the authors test the iELISA with some of the clinical samples?

Regarding the use of HPV-positive clinical samples to test our 7D2-based icELISA, we have performed determinations with HPV16 and HPV31-positive cervical samples, showing the specificity of our assay, given that no signal was detected (S3 Fig and S4 Table from the revised manuscript). The description of these assays can be found above in the response to Reviewer 1 - Question 3.

Overall the work done by Contreras et.al. is commendable and adds to the necessary information.

Reviewer #3: The manuscript titled “Novel Competitive Enzyme-Linked Immunosorbent Assay for the detection of the high-risk Human Papillomavirus 18 E6 oncoprotein” submitted to Plos one, describes a new ELISA technique developed by the authors Natalia E. Contreras, Julieta S. Roldán and Daniela S. Castillo. I appreciate the efforts undertaken by the authors in developing a novel ELISA based assay to detect E6 oncoprotein. It would be of diagnostic relevance if E6 oncoprotein could be detected directly from clinical specimen. I would like to recommend a couple of additional experiments (comments 1&2), an additional table (comment 3) and a few minor corrections (Comments 4-8), which I believe will substantially improve the quality of the article.

1) The test needs to be clinically validated before it can be recommended as screening tool. As a preliminary step authors could test their assay on at least a few clinically confirmed cytology positive/cervical cancer samples

To comply with the reviewer's request, we gathered clinical samples from women with a positive hybrid capture test. As we have mentioned before (see response to Reviewer 1 - Question 3), we tested our icELISA based on 7D2 mAb with these HPV16 and HPV31-positive clinical samples, confirming our assay's specificity as no signal was detected (S4 Table from the revised manuscript). 

2) HPV DNA PCR test is considered the gold standard test for HPV detection. I would recommend performing a comparative analysis of results from both PCR and this new ELISA test on confirmed cervical cancer biopsy samples. This would be highly informative as it would give out a direct comparison of accuracy, sensitivity and specificity of new method as against the current gold standard.

Since the cervical samples we obtained were from women with a positive hybrid capture test, we first had to determine the HR HPV genotype. For this purpose, we carried out PCR assays with the MY09-MY11 degenerate primers (which amplify the conserved L1 gene), followed by RFLP. According to the observed RFLP patterns, we determined the genotypes of the obtained samples to be HPV16 and HPV31 (S3 Fig from the revised manuscript).

3) Include a table with sensitivity and specificity of this new test.

Given that we obtained very few clinical samples, the results are not sufficient to build the required table.

4) Lines 265-267: Please restructure the sentence (“The standard curve for the …..” ) for better clarity. The sentence mentions “good correlation to the data”- which data are the authors referring to?

The sentence “The standard curve for the …..” has been rephrased for a better understanding (line 290-291 from the revised manuscript).

With respect to “good correlation to the data", we mean that our regression model fits the observed data values in our standard curve, given that we have obtained a coefficient of determination or R² equal to 0.99.

5) Supplementary table 3 needs to be restructured: The sequence identity matrix several duplicate values- eg: either the last 2 rows or last 2 columns could be deleted without loss of any information; even then the value 55.6 (HPV 18) remains repeated.

A sequence identity matrix has two entries from each sequence analyzed, and it is expected to contain identical halves. It is either correct to show the whole matrix or half of its information. We have modified our matrix to comply with the reviewer's request, showing only half of the identity scores recorded. This Table corresponds to S5 Table from the revised manuscript.

6) Line 102: Word “dialysis” misspelt

We have corrected the mistake (line 102 from the revised manuscript).

7) Line 131: “Tetramethylbenzidine” misspelt

We have corrected the mistake (line 131 from the revised manuscript).

8) Line 162: Word “subcloned” misspelt

We have corrected the mistake (line 162 from the revised manuscript).

---

## [Decision Letter · Decision Letter 1]

2 Aug 2023

Novel competitive enzyme-linked immunosorbent assay for the detection of the high-risk Human Papillomavirus 18 E6 oncoprotein

PONE-D-23-06213R1

Dear Dr. Castillo,

We’re pleased to inform you that your manuscript has been judged scientifically suitable for publication and will be formally accepted for publication once it meets all outstanding technical requirements.

Kind regards,

Arunava Roy, Ph.D.

Academic Editor

PLOS ONE

Additional Editor Comments (optional):

Reviewers' comments:

Reviewer's Responses to Questions

**Comments to the Author**

1. If the authors have adequately addressed your comments raised in a previous round of review and you feel that this manuscript is now acceptable for publication, you may indicate that here to bypass the “Comments to the Author” section, enter your conflict of interest statement in the “Confidential to Editor” section, and submit your "Accept" recommendation.

Reviewer #1: All comments have been addressed

Reviewer #2: All comments have been addressed

Reviewer #3: All comments have been addressed

2. Is the manuscript technically sound, and do the data support the conclusions?

Reviewer #1: Yes

Reviewer #2: Yes

Reviewer #3: Yes

3. Has the statistical analysis been performed appropriately and rigorously? 

Reviewer #1: Yes

Reviewer #2: Yes

Reviewer #3: Yes

4. Have the authors made all data underlying the findings in their manuscript fully available?

Reviewer #1: Yes

Reviewer #2: Yes

Reviewer #3: Yes

5. Is the manuscript presented in an intelligible fashion and written in standard English?

Reviewer #1: Yes

Reviewer #2: Yes

Reviewer #3: Yes

6. Review Comments to the Author

Reviewer #1: I understand the difficulty to get the exact HPV18 clinical sample, but still in the future it would be useful that you have positive data not just negative data to show your elisa assay work for positive clinical sample.

Reviewer #2: (No Response)

Reviewer #3: (No Response)

7. PLOS authors have the option to publish the peer review history of their article (what does this mean?). If published, this will include your full peer review and any attached files.

Reviewer #1: No

Reviewer #2: No

Reviewer #3: **Yes: **Priya R Prabhu

---

## [Editor Report · Acceptance letter]

7 Aug 2023

PONE-D-23-06213R1 

Novel competitive enzyme-linked immunosorbent assay for the detection of the high-risk Human Papillomavirus 18 E6 oncoprotein 

Dear Dr. Castillo:

I'm pleased to inform you that your manuscript has been deemed suitable for publication in PLOS ONE. Congratulations! Your manuscript is now with our production department. 

Kind regards, 

on behalf of

Dr. Arunava Roy 

Academic Editor

PLOS ONE